# Ultrasound Assessment of Larynx and Trachea in the Neonatal Period, Examination Standard with Predictive Values—Study Protocol

**DOI:** 10.3390/diagnostics13091578

**Published:** 2023-04-28

**Authors:** Łukasz Paprocki, Bartosz Migda, Renata Bokiniec

**Affiliations:** 1Department of Neonatology, Ujastek Medical Center, 31-752 Kraków, Poland; 2Diagnostic Ultrasound Lab, Department of Pediatric Radiology, Medical University of Warsaw, 02-091 Warsaw, Poland; 3Department of Neonatology and Intensive Care, Medical University of Warsaw, 02-091 Warsaw, Poland

**Keywords:** ultrasound, larynx, trachea, newborn, anatomy, technique, norms, high resolution

## Abstract

Diseases of the larynx and trachea are a heterogenous group of disorders. Their diagnosis frequently requires invasive methods. Ultrasound is a non-invasive, repeatable and safe diagnostic method, which has recently, thanks to the development of technology, provided for very accurate imaging of even small structures, as well as their assessment on dynamic examination. Ultrasound examination of the larynx and trachea will be performed in 2022–2023 in a group of randomly selected 300 stable neonates born between 32 and 42 weeks of gestation. The results of this study will be presented after data collection in accordance with the adopted methodology. To date, this will be the first study to describe the ultrasound anatomy of the larynx and trachea and to establish reference ranges for the size of individual structures of the larynx and trachea in the neonatal population. We expect that our study will contribute to the further development of this part of ultrasonography and will reduce the number of invasive procedures performed in the diagnostics of these organs in the future. This manuscript is a study protocol registered at ClinicalTrials.gov (Identifier NCT05636410) and approved by the Bioethics Committee of the Medical University of Warsaw (KB 65/A2022).

## 1. Introduction

Diseases of the larynx and trachea constitute a heterogenous group of disorders. They may include congenital anatomical disorders, neoplastic changes, vocal cord paralysis of varied aetiology or narrowing of the larynx associated with long-term intubation. The multitude of disorders of these organs necessitates the continuing search for diagnostic methods which will not only provide answers to clinical questions, but will also be safe and have the least level of interference with the wellbeing of the patient.

Diagnosis of diseases of the larynx and trachea with respect to neonates and infants is mainly based on endoscopy, magnetic resonance imaging and computer tomography. There are some challenges associated with these investigations which result in them being rarely used in everyday practice. One challenge concerns the invasiveness of the procedure during endoscopy of the upper respiratory tract or ionizing radiation in CT. Others include the not infrequent necessity of anaesthetizing the patient and the need to obtain special parental consent for the procedures, which is also not without significance. Moreover, the application of these techniques requires high numbers of highly qualified specialists for their performance and evaluation. Finally, an important constraint is the limited availability of fiberoscopy, MRI or CT in most neonatal departments, which results in the necessity of transporting the patient to another facility.

In recent decades, only a few studies have been reported which described the ultrasound anatomy of the larynx in pediatric populations, especially in diagnosis of infant laryngomalacia prior to and following intubation or that of the mobility of the vocal cords [1,2,3,4,5,6]. Some of the studies which involved small groups of subjects—intensive care unit patients—were performed using lower quality ultrasound devices or addressed only part of the larynx and thus did not provide for unequivocal conclusions to be drawn [6,7,8,9,10]. In the literature, there are also reports where ultrasonography is used to assess pathologies such as laryngeal cysts or paralysis of the vocal folds [11,12]. To date, no standards have been published concerning the size of the structures of the larynx and trachea or the mobility of the vocal cords on ultrasound examination in the neonate. Additionally, there are no studies describing the technique of ultrasound examination of these organs with respect to the neonates, which would cover all relevant aspects, starting from a comfortable position for the patient, through the assessment of individual levels of the larynx, and ending with the description of the norms of the size of the structures of the larynx and trachea based on a large group of neonates. Additionally, there are no recommendations which include ultrasound examination as a reliable component of the diagnosis of congenital disorders of the larynx or other diseases of this organ in the pediatric population. With regard to adult patients, in the literature we can find high-value descriptions of ultrasound anatomy of the larynx using high-resolution ultrasonography [13]. Ultrasound is a non-invasive, repeatable and safe diagnostic method, which has recently provided for very accurate imaging of even small structures. Furthermore, the easy availability of this examination may in the future contribute to the early diagnosis of diseases of the larynx and trachea without the need to prolong neonatal hospitalization, anaesthesia or transportation of the patient.

We plan to provide a complete ultrasound examination standard for the larynx and trachea. Initially, the study will describe the preparation of the office, the exact placement of the neonate and the examination technique that will least affect the patient’s comfort. Subsequently, the focus will be placed on determining the size and mobility standards of specific parts of the larynx and trachea in neonatal populations. Additionally, we plan to prepare an examination protocol of those organs to unify the results with other future studies. The expectation is that this protocol will allow clinicians to learn about laryngeal ultrasound and will allow other researchers to perform similar studies in other pediatric populations, as well as size and mobility standards of specific parts of these organs in the neonatal population. In addition, the use of neural networks to analyze the ultrasound images obtained will provide for the development of algorithms which could become an irreplaceable tool, not only in the diagnosis of disorders, but also in predicting disorders affecting further development of a newborn.

## 2. Experimental Design

The design is a prospective, multi-center cohort study. The trial protocol was approved by the Bioethics Committee of the Medical University of Warsaw on 10 October 2022 (decision number KB 65/A2022) and has been registered at ClinicalTrials.gov (Identifier-NCT05636410). Once the eligibility criteria are fulfilled and written consents are obtained, the newborns will undergo an ultrasound examination according to the established protocol. After examination, the captured images will be analyzed prospectively by researchers in a computer program to collect data on the dimensions of particular structures. Additionally, data obtained from each patient will be analyzed with the use of neural networks in order to develop artificial intelligence algorithms. The study design diagram is shown in Figure 1.

## 3. Materials and Equipment

Ultrasound examination of larynx and trachea will be performed in the Department of Neonatology and Neonatal Intensive Care at the Medical University of Warsaw and the Department of Neonatology at Ujastek Medical Center in Cracow, in 2022–2023, in a group of three-hundred randomly selected neonates, born between 32 and 42 weeks of gestation, who are stable and do not require intubation. The examination will take place in a specially prepared ultrasound room; therefore, only stable newborns who do not require intubation will be included. Prior to that, written consent will be obtained from at least one parent prior to the examination.

Inclusion criteria:Gestational age ≥32 weeks.

Exclusion criteria:Gestational age <32 weeks;Presence of significant congenital anomalies;Intubated neonate;Unstable neonate.

As recommended by WHO for similar research projects, a representative study group should include at least 200 study participants [14]. To improve the quality and reliability of the results, the number of participants for this study was set at 300. The calculated inter-observer and intra-observer concurrence (Kappa coefficient) will be calculated for all measurements.

The examination will be performed with high-class ultrasound machine (Philips Epiq 97G device, using the eL18-4 linear probe), using water-based, hypoallergic ultrasound gel (Aquasonic 100, Parker laboratories Inc., Fairfield, CT, USA), previously warmed to body temperature.

## 4. Details of the Procedure

### 4.1. Test Methods

Examination of the larynx and trachea will be carried out in the ultrasound room. To avoid neonate anxiety, windows will be shadowed and the ambient light will be kept low. Each time during examination, at least one parent will be able to accompany the child and make it feel comfortable. To avoid newborn anxiety, it will not be transferred to a special couch, but rather the ultrasound examination will be conducted in its crib. Prior to that, the researcher will make sure that the newborn has eaten between half and two hours before examination, and is fully calm or asleep. Right before examination, the baby will be sedated with a 30% oral glucose solution (Glux) and if a given neonate uses a pacifier, it will be removed. A small diaper roll will be placed under the newborn’s shoulder blades to ensure that the head is slightly tilted back to clearly show the structures of the neck. Each time the researcher will make sure that the newborn’s face and chest surface is parallel or nearly parallel so the test will not disturb its breathing (Figure A1). The study procedures will not affect neonatal and mother–infant care practices at the participating centers. The ultrasound examination will last up to 5 min or shorter in cases of newborn anxiety. At any time during ultrasound, the patient’s parent will have an opportunity to interrupt it without giving a reason. In such cases, the patient will be removed from the study.

The examination will be performed using a high-class ultrasound device (Philips Epiq 7G), with the eL18-4 linear probe, using water-based, bacteriostatic and hypoallergic ultrasound gel, previously warmed in the special ultrasound gel warmer to body temperature. In every possible case, the researcher will try to use the same settings, which include most of all the highest possible frequency (18 Mhz), examination depth set at 2.5 cm, ultrasonic signal gain set between 45 and 50%, dynamic range set at 55–60, focus zone range width of about 2 cm and placed on 1 cm depth. In cases in which the images obtained with respect to single patients, using these settings, prove to be of low quality, the researcher will change them to receive the best image quality and resolution.

All ultrasound data will be gathered by doctors with at least 6 years’ experience in US examinations. Every patient will receive an individual number and a three letter code (e.g., 000 XXX), that will allow the collection of anonymous data. In each case, at least six cine loops and several still images will be collected, according to the following plan. Every cine loop will have a duration of 10 s to provide for slow movement of the probe and the most advantageous visualization of the structures under examination. The researcher will exert as little pressure as possible with the ultrasound transducer in order to best visualize individual structures, so as not to disturb the patient’s comfort and not to change the anatomy of the examined structures. The first projection will show the transverse section of the medial part of the neck from the base of the tongue, through the larynx, up to the jugular notch of the sternum. This will be firstly made with a linear probe placed on the midline of the neck, with the probe angled 90 degrees to the skin and then slowly moved upward towards the tongue, and downward towards the jugular notch. During upward movement, the probe will be tilted, so that it remains at the same angle to the skin surface. This cine loop will show the general structure and position of the hyoid bone, larynx and trachea with their anatomical relationships. This image will also visualize supraglottis with preepiglottic space, as well as infrahyoid muscles on the level of hyoid bone. To obtain second and third bilateral recordings, the researcher will place the transducer parallel to each thyroid lamina angled 90 degrees to the skin and move it slowly up and down a few centimeters to show the thyroid level of the larynx. Those images (left and right sides) will show the transverse, lateral section at the level of the glottis in order to accurately visualize the laryngeal cartilages, vocal cords and structures posterior to the thyroid cartilage. Furthermore, this image will be used for the evaluation of the posterior part of the piriform recess, the interarytenoid muscles and the anatomical relationship between the larynx and esophagus. The fourth and fifth recordings will also be bilateral, obtained technically in the same way as the previous two cine loops, but only at the level of a glottis. Those images will show the transverse, lateral section applying the M-mode option, which allows for the accurate measurement of the amplitude and duration of the movement of the vocal cords. The sixth recording, also through the transverse section in the midline of the larynx, will be carried out by arranging a probe similarly to the first cine loop, but only at the glottal level. These images will document the movement and thickness of the vocal cords and the shape of the glottis while breathing. This view will also allow one to measure the thickness of the thyroid cartilage and the angle of its lamellae and to evaluate subglottic muscles. Recordings will be archived as DICOM files. After examination, the recordings will be saved to a dedicated, password-protected external drive for further processing of the images.

In every case, during further evaluation, the researcher who carried out the ultrasound examination and other researchers with at least 5 years’ experience in ultrasonography will take the following measurements using the dedicated software (Horos, Nimble Co. LLC d/b/a Purview, Annapolis, MD USA):Width of the larynx at the level of the glottis (measurement taken in the transverse plane from the external margins of the thyroid cartilage) (Figure A2);Depth of the larynx at the level of the glottis (measurement taken in the transverse view from the external margin of the thyroid cartilage to the external border of the annular cartilage) (Figure A2);Thickness of the thyroid cartilage in the midline and on both sides transversely at the level of the glottis (in the transverse plane measurement from external margins of thyroid cartilage and vertical measurement of thyroid cartilage above the anterior commissure at the level of well-visualized arytenoid cartilages in the transverse plane) (Figure A3);Angle formed by the lamina of the thyroid cartilage (measurement taken in the transverse plane with two lines drawn tangentially to the thyroid laminae) (Figure A4);Width of the glottis during calm breathing (measured in the transverse plane with lines drawn between two medial margins of arytenoid glands when they are farthest apart during breathing) (Figure A5);Angle formed by the vocal cords during calm breathing, considering the opening angle of individual chords (measurement taken in the transverse plane with a line drawn from posterior part of the anterior commissure to the medial margins of the arytenoid gland) (Figure A6);Surface area of the glottis during calm exhalation (in order to avoid distorting the result of measuring the glottal area using the outline of its visible borders, this measurement will be taken by calculating the area of a triangle, the basis of which will be the width of the glottis, and the height will be the depth of the glottis measured from the posterior part of the anterior commissure to the line drawn to measure glottis width, measurement taken in the transverse plane) (Figure A7);Thickness of the arytenoid muscle during inhalation and exhalation (measurement taken in the transverse plane in the midline from the outer surfaces of this muscle) (Figure A8);Thickness of the vocal cords in transverse projections, tangentially to the midline (measure in the transverse, lateral plane, ventral and tangentially to the arytenoid cartilages) (Figure A9);Amplitude and duration of vocal cord movement during breathing at rest (measurement taken in the transverse, lateral plane, using M-mode as the distance between two lines drawn by the movement of the outer surface of the vocal cord—lighter than the surrounding lines);Width and depth of the trachea (measurement taken in the transverse plane of the longest transverse dimension of the trachea from the external edges of the tracheal cartilages and measurement of the longest depth of the trachea from the external margin of the tracheal cartilage to the outer part of the membranous wall) (Figure A10);Width and depth of the tracheal lumen (measurement made in the transverse plane of the longest transverse dimension and longest depth of the hypoechoic area limited by the inner edges of the tracheal cartilages and membranous wall of the trachea) (Figure A11);Thickness and width of the supraglottis (measurement taken in the transverse plane of the thickest and the widest visible part of the supraglottis) (Figure A12);Thickness of the preepiglottic space (measurement taken in the transverse plane of the thickest part) (Figure A13);Thickness of the infrahyoid muscles at the level of the thyroid cartilage and hyoid bone (measurement taken in the transverse view of the thickest dimension between the outer edges of thyrohyoid and sternohyoid muscles as a group of infrahyoid muscles in the transverse plane, at the level of well-visible hyoid bone and glottis) (Figure A14).

In order to reduce the variability of the results, only measurements taken from patients who are breathing calmly will be considered. Crying, anxious or heavy breathing patients will be marked and will not be included in the measurements. These patients will not be excluded from the study to determine the percentage of patients who can be examined without compromising comfort in this population.

In addition to the data obtained from the ultrasound examination, the following patient information from the medical history will be collected: date and time of delivery, gestational age, age at the time of the study, Apgar score at the 1st, 5th and 10th minutes of life, gender, type of delivery, birth weight and length, weight on the day of the examination.

During the next stage of the study, the images acquired in the ultrasound examination will be analyzed with the use of neural networks in order to develop artificial intelligence algorithms. The method of these analyses will be determined at a later date, depending on the quality and quantity of information collected.

To date, there is no established standard technique for ultrasound examination of the larynx and trachea with respect to newborns. Furthermore, at the time of designing and commencing the study, there is no reference standard that would allow one to compare the obtained results.

### 4.2. Analysis

Statistical analysis will be conducted using R software, version 4.2.1. or later. Norms for measurements of the larynx and trachea will be defined with the following percentiles: 5th, 10th, 25th, 50th (median), 75th, 90th and 95th. Additionally, density charts will be drafted to visualize the distribution of data for each measurement. Normality of distribution will be assessed using the Shapiro–Wilk test and based on visual assessment of histograms. Differences in each measurement between subgroups (sex, Apgar score, delivery method) will be verified using t-test and ANOVA or their non-parametric equivalents, as appropriate. Correlation analysis will be conducted between measurements of the larynx and trachea and gestational age, age at the time of the study, Apgar score at the 1st, 5th and 10th minutes of life, gender, type of delivery, birth weight and length, and weight on the day of the examination. Additionally, regression analysis (univariate and multivariate) will be used to determine the relationship between the above anthropometric measurements and measurements of the larynx and trachea.

## 5. Expected Results

The main aim of the study is the accurate description of the ultrasound anatomy of the larynx and trachea, as well as the establishment of an examination standard for these organs in neonatal populations. The next objective is to set reference ranges for the size of individual structures of the larynx and trachea with respect to newborns. The study also aims to develop a universal method of ultrasound assessment of the mobility of the vocal cords on dynamic examination and to determine a rule for predicting the expected size of the laryngeal and tracheal structures in relation to anthropometric measurements.

Another purpose of the study is to promote the use of a non-invasive examination, such as ultrasonography, in the diagnosis of diseases of the larynx and trachea with respect to the youngest patients. Additionally, the study will be conducted to develop artificial intelligence algorithms based on artificial neural networks, which will allow one to make larynx and trachea diagnostics easier in the future.

## 6. Discussion

This study will be effected subject to the following limitations. The first is the restriction of the research group to the Polish population. In order to increase the research group and increase its heterogeneity, recruitment of patients will take place in two Polish neonatal centers. The second limitation is the lack of a valid ultrasound examination standard and reference size standards for the larynx and trachea in the study population, so the obtained images and measurements may differ significantly between individual researchers. To reduce this problem, each researcher will be trained in the technique of obtaining the required images before the study begins. Furthermore, each researcher will be instructed in identification and measurement of individual trachea and larynx structures. Additionally, high-class ultrasound machines will be used to acquire the highest quality images.

To avoid bias, the ultrasound will be performed in the presence of one researcher, who will then measure the individual structures of the larynx and trachea. The other researcher’s measurements will take place independently with no information about patient history and without any contact between the researchers.

To date, no detailed description of the ultrasound anatomy of the larynx and trachea in the newborn population has been published considering the norms of the measurements of individual structures of these organs. We hope that our study will contribute to the further development of this part of ultrasonography and will reduce the number of invasive procedures performed on newborns in the future.

## Figures and Tables

**Figure 1 diagnostics-13-01578-f001:**
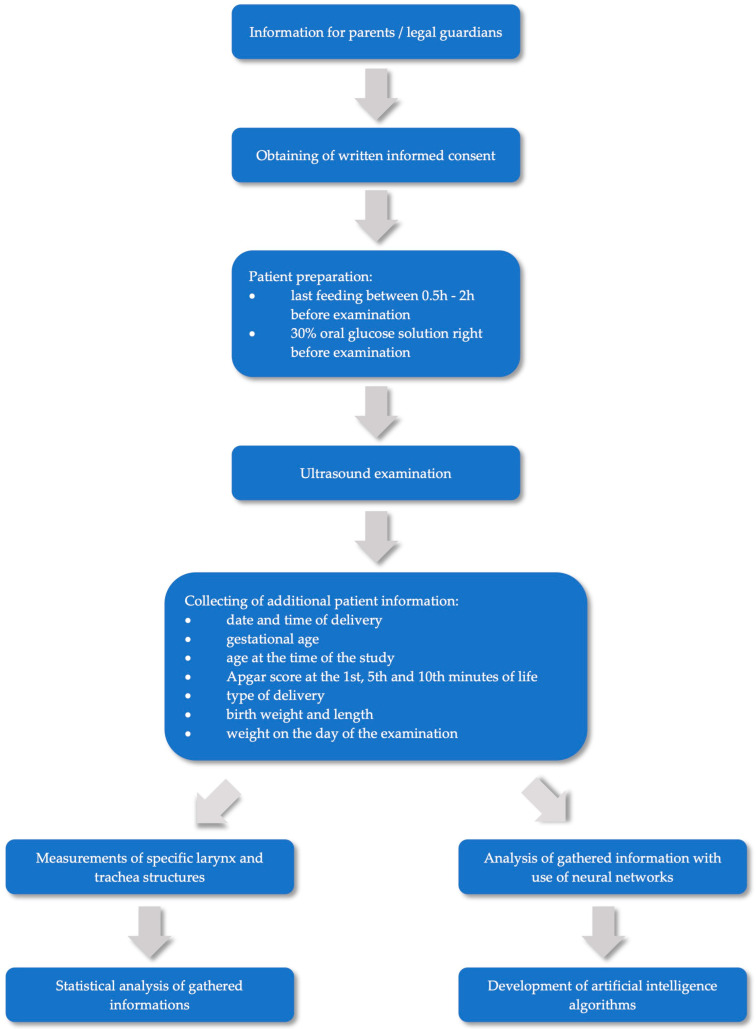
Study design diagram.

## Data Availability

This manuscript is a study protocol. No new data was created or analyzed in this study. Data sharing is not applicable to this article.

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
