# Peer review of "Ultrasound Assessment of Larynx and Trachea in the Neonatal Period, Examination Standard with Predictive Values—Study Protocol"

_diagnostics, 2023, doi:10.3390/diagnostics13091578_

Round 1
Reviewer 1 Report
Thanks for the opportunity to review this manuscript. This is a study protocol designed to establish standards doe neonatal ultrasound assessment of larynx and trachea. This is an interesting project. The manuscript needs some work to improve the language. I think it would be worth comparing with other imaging modalities such as CT or MRI to demonstrate if US is equally accurate. I am aware of studies in infants and children but not in neonates. Look forward to see the results of the study.
Author Response
Dear Reviewer,
I am very pleased to see that our study is interesting for you. Our study focuses on ultrasonography because we use it as the basic diagnostic tool in daily work.
Unfortunately, we do not have much experience in examining larynx and trachea in computed tomography or magnetic resonance imaging, so I am afraid that comparison you suggest if was made by our team could provide to inaccurate and erroneous conclusions. I believe that by conducting our study, it will be possible to increase interest in this topic in the medical community and will end up with further studies comparing these methods. We also made changes to the text, especially regarding its introduction. I hope these changes will give the reader more information about the purpose of our work and you will find it well.
Yours faithfully.
Reviewer 2 Report
In this study, the authors proposed an ultrasound assessment of the larynx and trachea using a state-of-art device. The expected results should be fascinating and useful.
Considering congenital and acquired subglottic stenosis is a significant condition for neonates, I suggest adding the measurements of the width and depth of the subglottic area as well as the thickness of the annular cartilage.
Author Response
Dear Reviewer,
I am very pleased to see that our study is interesting for you. In response to your suggestion to add additional measurements, I would like take notice to the technical difficulties associated with them. Due to the small distance between the glottis and the first thyroid cartilage, in our experience it was difficult to determine the exact place allowing for repeateable measurements of these structures. For this reason, in order to create the most reproducible larynx and trachea examination scheme, we decided not to include these structures in the examination. In our protocol, however, we have included the measurement of the trachea, which we intend to carry out in the closest possible place in relation to the glottis. I hope that such a measurement will be satisfactory and perhaps in the future it will allow to expand the diagnostic value of examining the subglottic part of larynx. We also made changes to the text, especially regarding its introduction. I hope these changes will give the reader more information about the purpose of our work and you will find it well.
Yours faithfully.
Reviewer 3 Report
Well planned study.
Best wishes
Author Response
Dear Reviewer,
I am very pleased to see that our study is interesting for you. We made some changes to the text, especially regarding its introduction. I hope these changes will give the reader more information about the purpose of our work and you will find it well.
Yours faithfully.